# Immunotherapies for the Treatment of Uveal Melanoma—History and Future

**DOI:** 10.3390/cancers11081048

**Published:** 2019-07-24

**Authors:** Timo E. Schank, Jessica C. Hassel

**Affiliations:** 1Department of Dermatology, University Hospital Heidelberg, 69120 Heidelberg, Germany; 2National Center for Tumor Diseases, University Hospital Heidelberg, 69120 Heidelberg, Germany

**Keywords:** uveal melanoma, immunotherapy, checkpoint inhibitors, IMCgp100, tebentafusp, dendritic-cell vaccination, adoptive T-cell therapy

## Abstract

Background: Uveal melanoma is the most common primary intraocular malignancy among adults. It is, nevertheless, a rare disease, with an incidence of approximately one case per 100,000 individuals per year in Europe. Approximately half of tumors will eventually metastasize, and the liver is the organ usually affected. No standard-of-care treatment exists for metastasized uveal melanoma. Chemotherapies or liver-directed treatments do not usually result in long-term tumor control. Immunotherapies are currently the most promising therapy option available. Methods: We reviewed both relevant recent literature on PubMed concerning the treatment of uveal melanoma with immunotherapies, and currently investigated drugs on ClinicalTrials.gov. Our own experiences with immune checkpoint blockers are included in a case series of 20 patients. Results: Because few clinical trials have been conducted for metastasized uveal melanoma, no definitive treatment strategy exists for this rare disease. The outcomes of most immunotherapies are poor, especially compared with cutaneous melanoma. However, encouraging results have been found for some very recently investigated agents such as the bispecific tebentafusp, for which a remarkably increased one-year overall survival rate, and similarly increased disease control rate, were observed in early phase studies. Conclusions: The treatment of metastatic uveal melanoma remains challenging, and almost all patients still die from the disease. Long-term responses might be achievable by means of new immunological strategies. Patients should therefore be referred to large medical centers where they can take part in controlled clinical studies.

## 1. Introduction

Uveal melanoma arises from melanocytes of the iris (3–5%), ciliary body (5–8%), or choroid (approximately 85%) [1,2]. It is the most common malignant primary intraocular tumor among adults. It is nevertheless a rare disease, with approximately one case per 100,000 people per year in Europe.

Although melanocytes are the cells of origin, the biology and clinical behavior of uveal melanoma are distinct from those of cutaneous melanoma. Mutations in the *GNAQ* or *GNA11* genes are often found in uveal [3,4] but not in cutaneous [5,6] melanoma. In contrast, *BRAF* and *NRAS* mutations frequently occur in melanomas of the skin [5,6] but are almost non-existent in uveal melanoma.

Primary uveal melanomas are usually either treated by surgery (enucleation or globe-preserving surgery) or radiation with Iodine-125 brachytherapy, resulting in similar survival outcomes [7]. Approximately 50% of all uveal melanoma patients will suffer from metastatic disease, predominantly to the liver. Due to the lack of lymphatic vessels in the choroid and alymphatic barrier of the sclera, primary uveal melanoma is widely protected from the immune system and spreads almost exclusively via the hematological route [8,9]. Moreover, the ability to metastasize depends on the tumor size and the genetic background of the uveal melanoma. Disease spread is associated with inactivating mutations in the tumor-suppressor gene *BRCA1*-associated protein 1 (BAP 1), which is present in 85% of metastatic uveal melanomas [10]. Furthermore, monosomy 3 is the main risk factor for metastases [11], and strongly correlates with decreased survival: The three-year overall survival rate among patients with monosomy 3 is 60%, whereas patients with disomy 3 have a three-year overall survival rate of 95–100% [12]. Once distant metastatic disease is established, the one-year overall survival rate is in the range 13–40%, with a median survival of two to eight months [13,14,15].

No standard-of-care treatment exists for metastasized uveal melanoma. Systemic chemotherapy does not have a compelling effect on overall survival [16]. Hepatic metastases are often the only detectable manifestation of stage-IV disease. Several local treatment options exist for this, including surgery, hepatic intra-arterial chemotherapy, transarterial chemoembolisation, isolated hepatic perfusion, selective-internal-radiation therapy (SIRT), and the percutaneous hepatic perfusion technique. Thus far, none of these strategies have provided a survival benefit for hepatic metastasized uveal melanoma patients in prospective, randomized studies [17]. This question is, however, currently under investigation in a trial comparing isolated hepatic perfusion with best alternative care (NCT01785316).

With regard to advanced cutaneous melanoma, newer systemic therapies have improved patient survival significantly. The field of immunotherapies in particular has yielded impressive results. These therapies aim to empower the immune system to induce anti-tumor immunity. Despite their differing clinical and biological behavior, uveal and cutaneous melanomas often share the same specific antigens (e.g., tyrosinase and gp100) [18,19]. Furthermore, several immune cells have been found in uveal melanoma, among these T-cells and dendritic cells [20,21,22], thus making metastasized uveal melanoma a possible target for immunotherapy. This review will focus on the different immunotherapeutic approaches studied so far, including one of the most promising: tebentafusp. 

We used the PubMed search function to select relevant recent studies on the treatment of uveal melanoma with immunotherapies. Key criteria for measuring the clinical effect of the treatments were incorporated into Table 1, including elevated lactate dehydrogenase (LDH) at baseline, therapy regimen, best response, (overall response rate (ORR) = complete response (CR) + partial response (PR)), median duration of response, (disease control rate (DCR) = CR + PR + stable disease (SD)), durable DCR (DCR ≥ 6 months), and median progression-free survival (PFS). If the outcomes were not clearly described in the manuscripts, these were obtained from tables within the publication. We included nine studies with a minimum of 10 patients per study. Furthermore, we retrospectively analyzed metastatic uveal melanoma patients treated with checkpoint inhibition from August 2014 until February 2019 at our own site. To obtain an overview of ongoing clinical studies of metastasized uveal melanoma, we used ClinicalTrials.gov to search for all uveal melanoma studies. We excluded trials that did not focus on advanced disease and those using non-immunotherapy-based regimens. Out of 141 uveal melanoma findings, we selected nine trials that matched our criteria. These are described in the following sections.

## 2. Dendritic-Cell Vaccination

Dendritic cells are antigen-presenting cells that can activate antigen-specific T-cells and therefore result in anti-tumor immune activity. This potential has been used to produce dendritic-cell vaccines for the treatment of stage-IV melanoma. Under this treatment, patients receive dendritic cells loaded with tumor-associated antigens (e.g., gp100, tyrosinase, or melanoma RNA). Recently a phase I/II study of advanced melanoma patients treated with dendritic-cell vaccination published a twelve-year follow-up. The authors come to the conclusion that the long-term survival of these patients is comparable with the long-term survival of advanced melanoma patients treated with Ipilimumab [23]. Along with these findings, in a study of 14 metastasized uveal melanoma patients treated with dendritic-cell vaccination, a DCR of approximately 70% was observed, as well as a durable DCR of approximately 20% (Table 1) and a median overall survival (OS) of 19 months [24]. No treatment-related adverse events (AEs) of grade 3 or 4 occurred. Observed AEs were grade-1 fatigue (five patients), flu-like symptoms (eight patients), and erythema at the intradermal injection site (six patients). However, best response was only SD. The main confounders of this study were its small sample size, non-randomized study design, and quite small number of patients with only a low tumor burden (elevated LDH in 21% of patients). These confounders might have resulted in a patient selection bias.

It is possible that a high tumor burden hampers the induction of an effective immune response due to the secretion of suppressive cytokines and activation of regulatory T-cells [25]. In this context, an open-label phase II study of 23 patients investigated the treatment of high-risk patients (identified by monosomy 3 of the primary tumor) with dendritic-cell vaccinations in an adjuvant setting after resection of the primary tumor. A three-year overall-survival rate of 79% was achieved and median disease-free survival (DFS) was 34.5 months [26]. Therapy was well tolerated: Transient flu-like symptoms (91% of patients) and erythema at the site of injection (87% of patients) were reported, but no grade 3 or 4 toxicities. A multicenter, open-label, randomized phase III study comparing adjuvant dendritic-cell vaccination with an observation group (observation only with staging every three months) is currently underway in an attempt to definitively determine whether dendritic-cell vaccination can prevent or delay progression among high-risk uveal melanoma patients (NCT01983748) [27].

## 3. Checkpoint Inhibitors

Checkpoint inhibitors are one of the most successful agents used in cutaneous melanoma therapy. Response rates among metastasized patients are in the range 40–50% for anti-programmed cell death protein 1 (PD-1) single-agent therapy (nivolumab or pembrolizumab) and up to 60% for combined immunotherapy with anti-cytotoxic T-lymphocyte-associated Protein 4 (CTLA-4) (ipilimumab) and anti-PD-1 (nivolumab) [28,29], thus rendering metastasized cutaneous melanoma a potentially curative disease. Expectations for its use as a treatment for uveal melanoma were consequently high. Unfortunately, the effect on uveal melanoma was not nearly as impressive as hoped for.

Zimmer et al. reported on a phase II trial of 53 patients with metastasized uveal melanoma who received ipilimumab (Table 1). Although most patients (85%) were pre-treated systemically, only 38% had an elevated LDH. The best achievable response was SD, with a DCR of 47% and a durable DCR of 21%. PFS was only 2.8 months, and OS was 6.8 months [30]. AEs were reported for 66% of patients. The most common of these were gastrointestinal disorders (diarrhea and colitis), skin-related toxicities (pruritus and rash), and hepatitis (increased transaminases). AEs of grade 3 or higher, mainly diarrhea and colitis, were seen among 36% of cases. When treated in accordance with guidelines, the AEs were usually reversible. However, one possible treatment-related death occurred as a result of pancytopenia with subsequent cerebral hemorrhage and respiratory insufficiency [30].

In a retrospective study of 96 patients, nivolumab monotherapy, pembrolizumab, and combined immunotherapy (ipilimumab + nivolumab) were given to 32, 54, and 15 patients, respectively. Most participants had not received prior treatment and were treated with PD-1 inhibitor monotherapy as first-line therapy. Approximately 50% of patients had an elevated LDH. Patients who received combined immunotherapy were significantly younger than those who received PD-1 monotherapy. For single-agent anti-PD-1 (nivolumab or pembrolizumab), an ORR of approximately 5% and PFS of approximately three months were observed [31]. For combined immunotherapy (ipilimumab and nivolumab), the ORR was approximately 17%, with a similar PFS of three months [31]. The DCR was approximately 20% for single-agent anti-PD-1 and 33% for combined immunotherapy. Two instances of PR occurred under combined immunotherapy and under nivolumab, and one PR under pembrolizumab. CR was not observed. Treatment-related AEs of grade 3 or higher (arthritis, autoimmune hepatitis, cardiac toxicity, and elevation of serum lipase) were recorded for four patients (7%) in the pembrolizumab group. Three patients (13%) in the nivolumab group reported AEs of grade 3 or higher (colitis, cardiac toxicity, arthralgia, and fatigue). One of these three patients died. In the combined immunotherapy group two patients (13%) experienced AEs of grade 3 or higher (hypophysitis, colitis, and thyroiditis). 

Another retrospective study of 56 patients treated with PD-1 or PD-L1 antibodies yielded similarly disappointing results. Here, an elevated LDH among approximately 70% of patients indicated an advanced tumor stage. Forty-eight patients (85.7%) had received prior systemic therapy, as is used for stage-IV disease. The ORR was 3.6% (two PR) and PFS was 2.6 months [32]. Moreover, seven patients (12.5%) reported grade-3 AEs (nausea, vomiting, hyperbilirubinemia, fatigue, colitis, arthralgia, and lymphopenia), and one patient discontinued treatment as a result (arthralgia). No AEs of grade 4 or 5 were noted.

At our own center, we retrospectively analyzed 20 patients who were treated with nivolumab, pembrolizumab, or the combination of ipilimumab and nivolumab. Patients were treated with checkpoint inhibition from August 2014 until February 2019. The data cut-off point was 28th February 2019. LDH was elevated among 55% of patients. The best response achieved was PR for two patients treated with pembrolizumab and ipilimumab plus nivolumab, respectively (11.8% of all evaluable patients, see Table 1, Figure 1). The ORR and DCR were 11.8% and 29.4%, respectively, with a durable DCR of only 5.9%. With a median PFS of only 2.75 months (two patients lost to follow-up, one patient ongoing) and a median duration of response of 2.85 months, the results of our own analysis are in agreement with the poor results seen in the studies discussed above. Furthermore, AEs of grade 3 or higher were present among 30% of all patients, resulting in the use of immunosuppressive cortisone treatment.

Recent data from an interim analysis of a prospective phase II, multicenter, open-label, single-arm study of the Spanish Melanoma Group were presented at the European Society for Medical Oncology (ESMO) 2018 congress (NCT02626962). Fifty metastasized uveal melanoma patients were treated with combined immunotherapy (nivolumab + ipilimumab). LDH was elevated among 32% of all cases, liver metastases were present among 76% of patients and extra-liver metastases among 56%. The ORR was 12%, and the DCR was 64%. PFS was 3.3 months, and the median OS was 12.7 months [33]. AEs of grade 3 or higher were reported for 27 patients (54%), including exanthemas, hepatitis, colitis, neurological deficit, anemia, thyroiditis, and hypophysitis, resulting in nine treatment discontinuations. These side effects were successfully controlled using the appropriate guideline treatment, with the exception of one case of acute thyroiditis, which resulted in death. Another prospective open-label trial investigating a combined immunotherapy with ipilimumab and nivolumab is currently underway (NCT01585194). 

Results were recently published from a phase I/II study of an adjuvant therapy with ipilimumab among uveal melanoma patients, including those at risk as defined by a high-risk molecular gene signature, monosomy 3, or apical thickness >8 mm on baseline echography. Ten patients were included, and distant metastasis-free survival after 36 months was assessed. Eight patients were still free of distant metastases after three years [34]. However, 90% of patients treated in the adjuvant setting had grade 3–4 toxicity, and one patient even suffered from blindness caused by temporal arteritis. Because of its small sample size, however, the value of this study remains unclear. The toxicity of immune checkpoint inhibition is, however, an important topic. The reporting of grade 3 and 4 side effects, and even death, illustrates the importance of weighing treatment benefits and risks against each other. The procurement of informed consent after a detailed explanation of possible side effects, including death, is essential. In this context, the additional benefit of ipilimumab in combined checkpoint inhibition for the treatment of metastasized uveal melanoma is uncertain. In particular, no references are available that confirm a prolonged PFS for this combined treatment, but it is known to lead to a greater number of AEs of grade 3 and higher. Consequently, combined immune checkpoint inhibition might not be a suitable treatment option for stage-IV uveal melanoma.

In summary, the success observed for checkpoint inhibitor treatment of metastasized cutaneous melanoma has not been observed for stage-IV uveal melanoma. Its limited efficacy for the treatment of advanced uveal melanoma could be due to several reasons. The liver is the main site of metastases, and it is known from cutaneous melanoma that liver metastases seem to respond less well than other types [35]. In addition, patients with a high tumor load—as indicated by an elevated LDH—are unlikely to respond [36]. In the retrospective analyses, and in our own case series, serum LDH activity was already increased among more than 50% of patients. In addition, uveal melanoma was shown to usually have a low mutational burden [37]. A high mutational burden is accompanied by an increased number of neoantigens, increasing the probability of recognition by immune cells [38]. Interestingly, patients with metastasized uveal melanoma and unusually high mutational loads have been successfully treated using checkpoint inhibition, experiencing prolonged survival or even complete remission [39,40,41]. In two of these cases, exome sequencing revealed a methyl-CpG-binding domain protein 4 (MBD4) loss-of-function mutation, resulting in a high mutational burden. Hence, an undetected high mutational load might explain the sporadic success of individual patients with checkpoint inhibitors. 

It could be shown that uveal melanoma metastases have only a few intra-metastatic CD8+ T-cells with high peritumoral predominance [42]. Thus, CD8+ cytotoxic T-cells seem to be excluded from their possible effector site. This problem might be solved by a new agent in the field of immunotherapies currently under investigation: The bispecific molecule tebentafusp. 

## 4. Bispecific Molecules 

Tebentafusp (also called IMCgp100) is one of a new class of anti-tumor reagents called immune-mobilizing monoclonal T-cell receptors against cancer (ImmTACs). Their mechanism is based on the assumption of a failure of the specific immune recognition and activation process mediated by the T-cell receptor, which results in a reduced anti-tumor effect. ImmTACs are bispecific molecules consisting of a tumor-epitope-directed T-cell receptor with high affinity and a CD3-specific, humanized single-chain antibody fragment. Thus, they aim to redirect T-cells to cancer cells and activate them, resulting in tumor-cell lysis [43]. 

The T-cell receptor domain of tebentafusp binds exclusively to an human leukocyte antigen (HLA)-0201-restricted gp100 peptide, which is expressed on melanoma tumors such as cutaneous and uveal melanomas [18,19]. Tebentafusp overcomes the peritumoral restriction of T-cells and results in T-cell trafficking in the tumor. Moreover, a notable upregulation of PD-1 and PD-L1 occurs within the tumor [44]. In the first in-human study, presented at the American Society of Clinical Oncology (ASCO) Annual Meeting 2016 [45], tebentafusp was investigated in relation to both uveal melanoma (14 patients) and cutaneous melanoma (33 patients). With regard to cutaneous melanoma, a PR was seen for two patients (6%) and SD for nine patients (27%), whereas disease progression and ongoing DCR were recorded for 20 patients (61%) and seven patients (21%), respectively, after 16 weeks. With regard to uveal melanoma, two instances of PR (14%), eight of SD (57%), and four of PD (29%) were observed, leading to a DCR and ongoing DCR of 71% and 57%, respectively, after 16 weeks. An analysis of the subsequent phase I study for metastasized uveal melanoma included 19 heavily pre-treated patients (Table 1), 73% of whom had an elevated LDH. Treatment with tebentafusp led to two instances of PR (11%) and another DCR of 71%, with a durable DCR of 41%. PFS was stated at 24.3 weeks. One-year overall survival was as high as 74%, and the median OS had not yet been met. The analysis also revealed that patients had been treated with various agents after disease progression, including PD-1 and CTLA-4 inhibition, whereupon a PR was recorded for two patients under combined immunotherapy [46,47]. This suggests an ongoing treatment effect.

No deaths and no treatment discontinuations as a result of AEs were reported during this phase I study. However, all participants (100%) experienced AEs, and 15 patients (79%) experienced AEs of grade 3 or higher (79%). The most common AEs of grade 3 and 4 were liver-enzyme elevation, erythema, and hypotension (16% in each case) [44]. Interestingly, AEs ceased over the course of treatment, suggesting a habituation effect.

As the results for melanoma have been promising, the early stages of investigation are also underway for therapy with ImmTAC molecules for several other cancer entities, for example, those of the bladder, oesophagus, stomach, head and neck, female reproductive system, and lungs [48]. Furthermore, early research on immune-mobilizing monoclonal T-Cell receptors (TCRs) against viruses or bacteria (ImmTAV™, ImmTAB™, Immunocore Ltd, Oxfordshire, United Kingdom) has been initiated [49].

## 5. Adoptive T-Cell therapy

Adoptive T-cell therapy using expanded autologous tumor-infiltrating lymphocytes (TIL) has been described as inducing salvage responses in several refractory solid tumors [50,51]. This complex therapy consists of several steps. First, a metastasectomy is performed on the patient to procure tumor tissue and to extract autologous TILs. Ex vivo, these lymphocytes undergo large-scale expansion. Patients then receive lymphodepleting chemotherapy followed by intravenous infusion of the lymphocytes and interleukin-2 doses. 

In an interim analysis of a phase II trial of 20 pre-treated patients with metastatic uveal melanoma using TIL-therapy, 52% had an elevated LDH. Best response was one CR, which was still ongoing at the time of interim analysis after 21 months. The ORR was approximately 35%, and the DCR approximately 43%, with a durable DCR of approximately 10% (Table 1). The median duration of response during interim analysis was at least four months (ongoing) [52]. However, chemotherapy-induced bone marrow suppression entails the risk of life-threatening infections. Moreover, all patients (100%) suffered from AEs of grade 3 or higher due to chemotherapy. One patient died from severe treatment-refractory respiratory syncytial virus pneumonia.

To potentially increase efficacy and be independent of tumor material, genetically modified autologous T-cell receptors are investigated in ongoing clinical trials, e.g., against preferentially expressed antigen in melanoma (PRAME) (NCT02743611) or MART-1 (NCT02654821). A trial with Ny-Eso TCR therapy in patients with melanoma was closed early because of a high mortality (NCT01350401). However, another pilot trial combination of transgenic Ny-Eso TCR therapy with dentritic cell vaccination with or without ipilimumab was less toxic, but the two included melanoma patients did not benefit from the treatment [53].

Chimeric Antigen Receptor (CAR)-T-Cell therapy is another promising adoptive T-cell therapy strategy. Hereby, via apheresis, isolated peripheral T-cells become further processed by transfection with genomic products, leading to an expression of transmembrane receptors targeting a tumor-specific cell surface protein [54]. This therapy modality showed impressive results in children with acute B lymphoblastic leukemia that had failed all other conventional therapies [55]. Currently, there are early CAR-T-Cell therapy studies for stage-IV cutaneous melanoma—but not for uveal melanoma—recruiting, targeting among others the cell surface proteins cluster of differentiation 20 (CD20) and disialoganglioside (GD2). However, in solid tumors similar results as mentioned above for leukemia [55] have not yet been demonstrated. Hypotheses referring to this include that the tumor microenvironment blocks the CAR-T-Cell effect, that the CAR-T-Cells become exhausted before they can eradicate the tumor, or that the focused on antigen is not uniformly expressed on the tumor cell surfaces. Nonetheless, a recent publication could demonstrate that CAR-T-Cells in a humanized mouse model can kill uveal and cutaneous melanoma cells in vitro and in vivo [56], suggesting that this therapy approach might still be successful. In this respect, T-Cell receptor therapy might be a more suitable alternative for uveal melanoma as this therapy aims for intracellular tissue differentiation antigens expressed via the major histocompatibility complex (MHC)-peptide, as for example gp100 or Melanoma antigen recognised by T-cells 1 (MART-1) which are highly expressed in uveal melanoma cells [18]. A phase I/II study (NCT02654821) is currently investigating the effects of this therapy in melanoma (cutaneous, uveal, mucosal melanoma, and melanoma of unknown primary).

## 6. Other drugs on the horizon

Integrated as a fundamental component of oncology, immunotherapies are one of the most effective agents for the treatment of many cancer entities, including cutaneous melanoma. Although treatment algorithms cannot simply be transferred to uveal melanoma, each of the above-mentioned immunotherapeutic strategies nonetheless seem to offer beneficial potential by empowering the immune system to control metastasized uveal melanoma. However, the already well-established PD-1 or CTLA-4 inhibitors in cutaneous melanoma are not nearly as successful in uveal melanoma. Hence, immunotherapeutic research efforts are ongoing to combine them with different reagents or therapy modalities in the hope of achieving cumulative effects. Several studies have therefore been initiated that combine checkpoint inhibition with local therapies such as SIRT (NCT02913417), intra-metastatic injection of oncolytic viruses combined with external-beam radiation (NCT02831933), or intravenous application of oncolytic viruses (NCT03408587). The combination of pembrolizumab with the histone-deacetylase inhibitor entinostat is also being investigated (NCT02697630). Targeting other checkpoint molecules that have arisen interest in cancer research, as for example Lymphocyte-activation gene 3 (LAG-3), T-cell immunoglobulin and mucin domain 3 (TIM-3) or T-cell immunoreceptor with immunoglobulin (Ig) and Immunoreceptor tyrosine-based inhibitory motif (ITIM) domains (TIGIT), has not yet found its way into clinical uveal melanoma trials. They were found to be induced in T-cells when treated with PD-1 inhibitor therapy [57,58], and therefore may play an important role in tumor cell escape mechanisms under PD-1 therapy. In fact, coinhibition of PD-1 and LAG-3 showed clinical activity in patients with previously treated metastatic or unresectable melanoma whose disease progressed during prior anti–PD(L)1 therapy [59]. 

One of the variables most likely to influence the effect of immunotherapy is the number of somatic mutations within the tumor. Uveal melanoma shows generally a low number of tumor neo-antigens when compared to cutaneous melanoma. Hence, more personalized treatment approaches might help to overcome this obstacle. Four molecularly distinct subgroups of uveal melanoma have recently been found, connecting genomic analysis with clinical outcome. Knowledge of these subgroups will enable further personalized risk stratification of patients in future [37]. In this context, a recently published first-in-human study in advanced cutaneous melanoma patients treated with personalized RNA vaccines demonstrated objective responses, and even one complete response in conjunction with PD-1 inhibition [60], showing a possible treatment approach also for uveal melanoma. A notable response has been observed in an early phase study of adoptive T-cell therapy for metastasized uveal melanoma [52]. However, the final results of this TIL-trial have not yet been published. If efficacy can be further increased by specific TCR or CAR-T-Cell therapies is still speculative, clinical trials are ongoing (see above). 

With a PFS of approximately six months and a durable DCR of 41%, the potential of tebentafusp as a single agent for the treatment of uveal melanoma has been shown. It presents currently the first gleam of hope for metastasized uveal melanoma patients. If possible, it should be provided to patients suffering stage-IV disease. To date, it shows the best evidence when compared to alternative immunotherapeutic treatments (Figure 2). However, at present the use of tebentafusp is restricted to patients with HLA-0201 positivity, which is expressed in approximately 50% of the Caucasian population. If efficacy is confirmed in the ongoing clinical trials an expansion to other HLA-types is warranted. Furthermore, tumor shrinkage seems quite rare, whereas stable disease is more often achieved. This gives rise to new ideas about possible combinations with other therapeutic agents such as PD-1 inhibitor-based regimens, adoptive T-cell therapy, dendritic T-cell vaccination, or targeted therapies. These approaches all aim to empower the immune system and thereby elicit a possible increased immune response. Further investigation by means of randomized prospective trials is therefore needed. Because uveal melanoma is such a rare disease, however, sample size remains one of the most substantial limitations of all studies. 

## 7. Conclusions

Immunotherapies are one of the most exciting and successful oncological treatment strategies, yielding encouraging results even for stage-IV uveal melanoma. However, the treatment of metastasized uveal melanoma remains challenging, and further investigation is needed. Patients should hence be encouraged to participate in clinical study treatments.

## Figures and Tables

**Figure 1 cancers-11-01048-f001:**
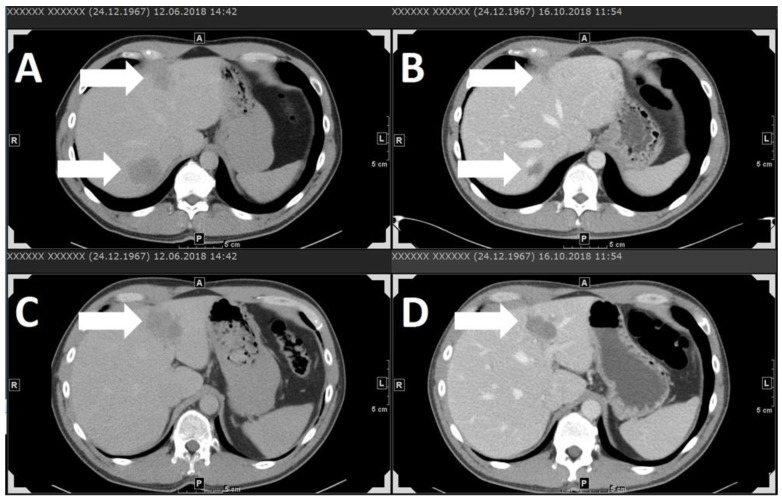
Stage-IV uveal melanoma patient with liver metastases shown in contrast-agent-enhanced computer tomography (CT) scans (**A**–**D**). Partial response of liver metastases under combined immune therapy with ipilimumab 3 mg/kg and nivolumab 1 mg/kg every three weeks. A and C show liver metastases before immune therapy. B and D show liver metastases in the same location (A corresponds to B, C corresponds to D) after three administration cycles. Arrows indicate metastases. With thanks to O. Sedlaczek, Department of Diagnostic and Interventional Radiology, Heidelberg University Hospital**.**

**Figure 2 cancers-11-01048-f002:**
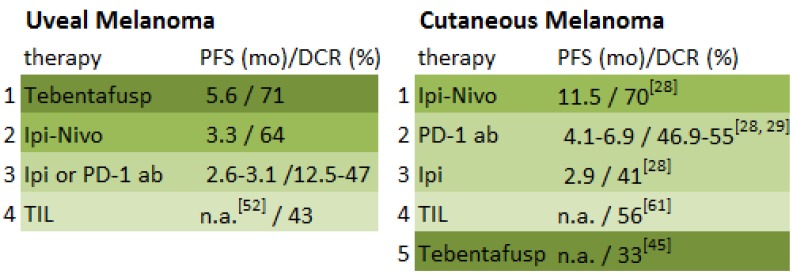
Overview of best evidence to date for immunotherapeutic treatment in uveal melanoma compared to cutaneous melanoma [61].

**Table 1 cancers-11-01048-t001:** Summary of immunotherapy studies for uveal melanoma (UM).

Study	Study Design	Number of Patients	Elevated LDH at Baseline	Therapy	Best Response	Overall Response Rate (CR + PR)	Median Duration of Response (mo)	Disease-Control Rate (CR + PR + SD)	Durable Disease-Control Rate (CR + PR + SD ≥ 6 mo)	Median PFS/DFS (mo)	AEs Grade ≥3 %
[24]	case series, stage-IV UM	14	21%	dendritic-cell vaccination	SD	0%	n.a.	71%	21%	n.a.	0
[26]	adjuvant, open-label, phase-II study, UM	23	n.a.	adjuvant dendritic-cell vaccination	n.a.	n.a.	n.a.	n.a.	n.a.	34.5	0
[30]	multicentre, phase-II, stage-IV UM	53	38%	ipi 3 mg/kg Q3W	SD	0%	n.a.	47%	21%	2.8	36%, one death
[31]	retrospective, multicentre, stage-IV UM	pembro: 54	57%	pembro 2 mg/kg Q3W	PR	4.7%	n.a.	22.7%	n.a.	3.1	7%
		nivo: 32	53%	nivo 3 mg/kg Q2W	PR	4.7%	n.a.	18.7%	n.a.	2.8	13%, one death
		ipi + PD-1 ab: 15 total: 86	47%	ipi 3 mg/kg + nivo 1 mg/kg Q3W, followed by nivo 3 mg/kg Q2W ipi 1 mg/kg + pembro 2 mg/kg Q3W, followed by pembro 2 mg/kg Q3W ipi 1 mg/kg + nivo 3 mg/kg Q3W, followed by nivo 3 mg/kg Q2W	PR	16.7%	n.a.	33.4%	n.a.	2.8	13%
[32]	retrospective, multicenter, stage-IV UM	pembro: 38 nivo: 16 atezo: 2 total: 56	71%	pembro 2 mg/kg Q3W 10 mg/kg Q2W 10 mg/kg Q3W Unknown Q3W nivo 1 mg/kg Q2W 2 mg/kg Q2W 3 mg/kg Q2W 10 mg/kg Q2W atezo 10 mg/kg Q2W 15 mg/kg Q2W	PR	3.6%	n.a.	12.5%	8.9%	2.6	13%
Case series HD, 2019 (unpublished)	retrospective, monocentre, stage-IV UM	pembro: 12 nivo: 1 ipi + nivo: 7 total: 20	55%	pembro 2 mg/kg Q3W nivo 3 mg/kg Q2W ipi 3 mg/kg + nivo 1 mg/kg Q3W, followed by nivo 3 mg/kg Q2W	PR	11.8%	2.85	29.4%	5.9%	2.75	30%
[33]	prospective, phase-II, multicentre, open-label, single-arm, stage-IV UM	50	32%	ipi 3 mg/kg + nivo 1 mg/kg Q3W, followed by nivo 3 mg/kg Q2W	PR	12%	n.a.	64%	n.a.	3.3	54%, one death
[46,47]	phase-I study, prospective, stage-IV UM	19	73%	tebentafusp (IMCgp100)	PR	11%	7.1	71%	41%	5.6	79%
[52]	interim analysis of a monocentre, two-stage, single-arm, phase-II study, stage-IV UM	21	52%	adoptive T-cell therapy	CR	35%	4+	43%	10%	n.a.	100%, one death

LDH, lactate dehydrogenase; UM, uveal melanoma; pembro, pembrolizumab; nivo, nivolumab; ipi, ipilimumab; atezo, atezolizumab; PD-1 ab, PD-1 antibody; CR, complete response; PR, partial response; SD, stable disease; HD, National Center for Tumor Diseases, University Hospital Heidelberg; n.a., not assessed; Q2W, every two weeks; Q3W, every three weeks; PFS, progression-free survival; DFS, median disease-free survival; mo, months; AEs, adverse events; death, treatment-related death.

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
