# Peer review of "Immunotherapies for the Treatment of Uveal Melanoma—History and Future"

_cancers, 2019, doi:10.3390/cancers11081048_

Round 1
Reviewer 1 Report
The topic of the article is very actual and interesting. The article is well structured, the methods adequately described.
The writing is clear. English language is adequate and correct.
The figure is of good quality.
I suggest the Authors to integrate the Table S1 into the main text.
Reviewer 2 Report
Schank and Hassel give a nice overview in some of current uveal melanoma immunotherapy trials. However, some of the references cited in the manuscript are too old or other important ones are missing and thus, it does not match the current data and knowledge in some aspects.
As this is a review article on clinical trials, section 2 of the manuscript seems to be redundant to me and can be taken out. They could include one or two sentences in the introduction section of the manuscript.
To date, the biology of DCs and its immunogenic role are better understood. As a result, DC-based vaccination for mutants and neoantigens has shown to be a reasonable treatment option for cutaneous melanoma patients, which was recently reported in prestigious journals (Sahin 2017 doi: 10.1038/nature23003; Schuler 2017 doi: 10.1172/jci.insight.91438). The authors must comment on that and include it in their article.
Also, I would take the old reference out, comparing DC-vaccines to Dacarbazine, as it does not reflect the current data any more. The authors should also discuss, if the vaccination against tumor neoantigens could also be an option for uveal melanoma patients.
Could the author also comment on other checkpoint molecules besides PD1 and CTLA4 (Lag3, Tim3, Tigit, ect.)? Are there clinical trials on these agents in uveal melanoma? Could these agents be more promising than PD1/CTLA4? If not, why?
Regarding ACT, the authors must absolutely also comment on CAR T cells, which -similar to checkpoint inhibition in melanoma- has shown impressive response rates for lymphomas and leukemias, and which is now also approved for these malignancies. I know that there are clinical CAR T trials on cutaneous melanoma currently running in the USA at U-Penn (Carl June) and at NIH, but also in Germany (e.g. CD20 CAR T at Univ. of Cologne). What do the authors think of CAR Ts in uveal melanoma?
And what do the authors think about TCR-T cell therapy in uveal melanoma? TCR T cells targeting MART1, for instance, showed promising results back 10 years ago. Could this be a treatment option?
I miss a little bit of a „vision“, „inspiration“, or a potentail „treatment strategy“ by the authors throughout the manuscript (independent of the current clinical trial results). What do they think will most likely help in the future? Is it only IMCgp100? Combinational therapeutic strategies? Is there any hope at all for these patients?
Reviewer 3 Report
This review focusses in uveal melanoma and the putative immunotherapies available to treat metastatic uveal melanoma.
The review analyzes all ongoing trials on this subject and leads to interesting conclusions, however, it would benefit from a graph or image that summarize all putative immunotherapeutic treatments for metastatic uveal melanoma and the differences with cutaneous melanoma. One or two images would improve this review.
Round 2
Reviewer 2 Report
The authors reponded well to the comments. However, some english changes are required prior to publication (as one example: in line 283 is should be "are" instead of "become" --> became is a so-called false-friend in German language).